# Advances in Optical Based Turbidity Sensing Using LED Photometry (PEDD)

**DOI:** 10.3390/s22010254

**Published:** 2021-12-30

**Authors:** Cormac D. Fay, Andrew Nattestad

**Affiliations:** 1SMART Infrastructure Facility, Engineering and Information Sciences, University of Wollongong, Wollongong, NSW 2522, Australia; 2School of Chemistry, Monash University, Clayton, VIC 3800, Australia; anattest@uow.edu.au; 3Intelligent Polymer Research Institute, AIIM Facility, University of Wollongong, Wollongong, NSW 2522, Australia

**Keywords:** turbidity, NTU, water, quality, PEDD, photodiode, low-power, ISO 7027, sensitivity, LOD, low-cost

## Abstract

Turbidity is one of the primary metrics to determine water quality in terms of health and environmental concerns, however analysis typically takes place in centralized facilities, with samples periodically collected and transported there. Large scale autonomous deployments (WSNs) are impeded by both initial and per measurement costs. In this study we employ a Paired Emitter-Detector Diode (PEDD) technique to quantitatively measure turbidity using analytical grade calibration standards. Our PEDD approach compares favorably against more conventional photodiode-LED arrangements in terms of spectral sensitivity, cost, power use, sensitivity, limit of detection, and physical arrangement as per the ISO 7027 turbidity sensing standard. The findings show that the PEDD technique was superior in all aforementioned aspects. It is therefore more ideal for low-cost, low-power, IoT deployed sensors. The significance of these findings can lead to environmental deployments that greatly lower the device and per-measurement costs.

## 1. Introduction

Turbidity is a well-established and important measure of water quality, describing the degree of ‘cloudiness’ caused by suspended particles (colloids) [1]. In the field of environmental monitoring, increases in turbidity in bodies of water such as rivers or lakes is often evidence of the impact of human activities such as construction, mining or agriculture [2,3]. Typically stormwater runoff from such sources results in increased concentrations of suspended particles, which can directly absorb sunlight leading to increased water temperatures. Such suspensions can also decrease the amount of light available for sub-surface photosynthetic reactions to take place due to either competitive absorption or reflection. Both the above issues have negative impacts on oxygen content, leading to further impacts on the surrounding ecosystem [1].

Quantifying the turbidity of a water source, both in terms of environmental and/or health concerns, is of high importance, with high speed, reliable measurements urgently needed [4]. At present, batch sampling is typical, with these taken from a restricted number of locations, due to the inherent limitations of manual grab sampling, before being transported to centralised facilities, equipped with sophisticated instrumentation and tested by highly trained personnel [5,6]. Although this provides high measurement precision and accuracy, it poses limits in terms of scaling and restricts the amount of data that can be obtained regarding temporal and spatial variations [5,7]. In recent years, some commercial solutions have emerged, capable of performing in situ measurements, such as the OBS501 from Campbell Scientific, ITM-51 from Anderson-Negele, ZWQ-TUB1 from Zeta and EXO3 from YSI. A detailed review of these and others has been provided by Abdulai et al. [8]. These are, however industrial grade systems, costing ca. >$10 k per unit for the loggers alone, with ad hoc and closed source communications, telemetry, and data access representing additional costs. While robust and offering options for in situ monitoring, the cost-base continues to impede widespread uptake of large scale monitoring programmes and as such the research community continues to investigate low-cost alternatives [9,10,11]. According to Popek et al. [12] a standard turbidity meter has a typical range of 0–1000 NTU (Nephelometric Turbidity Unit) and an accuracy of ±2–3%. This is supported by ranges reported on governing websites such as the UK Water Quality archive (environment.data.gov.uk/water-quality, accessed on 5 December 2021) in addition to those reported in the literature [13,14] where a typical range of 0–100 NTU is most likely, but allowing a scope of 0–1000 NTU for detection of events outside the nominal range [15,16].

The analysis of Wang et al. [14] suggests that low-cost turbidity sensing has only recently become viable due to the use of sensors that are orders of magnitude more lower in cost than the previous commercially available options. As part of their work they examined a low-cost, off-the-shelf, commercially available turbidity sensor (SEN0189, Gravity—designed for white-goods), noting that it suffered from accuracy, sensitivity and power issues, rendering them unsuitable for providing reliable in situ measurements, with this view also supported by Hakim et al. [17]. In place of the commercially available unit, Wang et al. [14] proposed a turbidity sensor based on an LED and photodiode (PD). Similarly, other works using a PD for turbidity sensing have been reported [13,18,19,20,21,22]. However, the use of an LED as the light detector for turbidity sensing remains unexplored.

For almost five decades, LEDs have been used as light detectors, since first studied by Mims in 1973 [23]. Due to the lack of sensitivity and low photocurrent [24], few individuals implemented LEDs as sensors until in 2004 when Lau et al. [25] demonstrated their use as an analytical detector in an unconventional manner. Specifically, they used the inherent capacitance of the diode (typically in the picofarad range) under reverse bias in a charge-discharge process, where the discharge time is proportional to the incident light intensity (assuming the same spectrum) and can be determined using a microcontroller’s I/O pin. The term Paired Emitter Detector Diode (PEDD) was coined to describe this operation [25,26]. Since then, the PEDD approach has been successfully used in analytical sensors in the solution phase (ammonia [27], nitrite [28], phosphate [29]), gas phase (acidic vapor [30], carbon dioxide [31], oxygen [32]), as well as in biological applications (hemoglobin [33], sweat constituents [34,35,36], or saliva [37]). Despite its success in these other areas, the PEDD technique is almost entirely unexplored for analytical measurement of turbidity.

The study we present here examines the quantitative detection/measurement of turbidity employing the charge/discharge PEDD detection technique and conforming to the ISO 7027 standard [38]. As discussed above, the standard implementation consists of a photodiode and an LED in the context of low-power deployed sensors. Consequently, the PEDD setup is examined and compared to the PD-LED setup in terms of cost, spectral sensitivity, physical arrangement (transmission and scattering), power use, sensitivity and limits of detection. This takes place under laboratory settings and examines turbidity values in the 0–100 NTU and also in the 0–1000 NTU ranges for reasons discussed previously.

## 2. Materials and Methods

### 2.1. Materials and Preparation

Based on reasons noted previously in the literature [12,39] as well as from data provided by governing repositories, higher (1000 NTU) and lower (100 NTU) ranges were considered. To directly reflect this, turbidity calibration standards were sourced (TURBP1000 and TURBP100, Sigma Aldrich, Burlington, MA, USA) and used either as purchased or diluted to various concentrations using purified water (Milli Q, Burlington, MA, USA). Both standards were kept refrigerated upon delivery. To prepare the suspensions, the calibration standards were removed from storage (refrigeration) and stirred to ensure that no settling took place while stored and result in homogeneous suspensions. The 1000 and 100 NTU standards were diluted to produce turbidity suspensions (within their respective range) in steps of 200 and 20 NTU, respectively. These will be denoted henceforth as the high (1000 NTU) and low (100 NTU) concentration ranges.

### 2.2. Light Components and Spectral Measurements

Appropriate emitters and detectors were sourced in order to comply with the ISO 7027 turbidity measurement standard [38]. An OSRAM SFH 4550 (OSRAM Opto Semiconductors, Regensburg, Germany) LED was sourced as an emitter, as it has emission centered at 860 nm (the midpoint of the ISO recommended wavelength range), a high radiant intensity 2000 mW/sr capacity, and a narrow emission angle of ±3°. Two Silicon photodiodes (PD) were sourced, SFH 213 and SFH 213 FA (OSRAM Opto Semiconductors), with peak sensitivities in LED emission range (850–900 nm, noting that this is also within the ISO standard). All diodes (LED and PD) had the same packaging size (5 mm T-1 3/4), and the casing of the SFH 213 FA PD was coloured to filter visible light, making it more robust to stray visible light than its SFH 213 FA counterpart.

The PD and LED were characterised as photovoltaic devices with both current-voltage (J-V) sweeps under white light using a Solar Simulator (PV Measurements, Boulder, CO, USA) and External Quantum Efficiency (EQE; QEX10, PV Measurements, Boulder, CO, USA) instruments. EQEs were measured under monochromatic illumination at wavelengths ranging from 300 to 1100 nm, stepped in 10 nm increments and under short circuit conditions. For reference, the emission spectra of the LED was obtained from the manufacturer’s datasheet and through the use of open source software (WebPlotDigitizer v4.3, automeris.io/WebPlotDigitizer, accessed on 5 December 2021) [40,41].

### 2.3. Sensing Arrangement

To reflect the ISO 7027 method of analytical measurements of turbidity [38] and for similarities to a spectrophotometer system, disposable, low volume, polystyrene cuvettes of 1 cm path length were sourced (Labtek, 650.200.110). Figure 1 shows the experimental design, including the use of the in house designed light chamber accommodating the emitter LED, two detector LEDs as well as a photodiode detector with the detector and emitter facing each other and an orthogonal arrangement with the detector and emitter placed orthogonal/normal to each other, again conforming with the ISO standard. Note that a similar approach was used by Lau et al. [42]. The four (4) mounting points were designed to firmly hold standard 5 mm emitters or detectors and also to control the light arrangements. The chamber, presented in a grey colour, was fabricated using black PLA on a Flashforge Dreamer FFF 3D printer.

### 2.4. Emitter and Detector Implementation

Figure 1 presents the electrical schematic showing the arrangement of the components required for the light emitter and detectors drawn using CAD software (KiCAD, 5.1.7). The emitter LED was issued with two resistors in series. The protective resistor ($R_{p}$) was set at 56Ω to protect the LED from over-current damage while using a 5 V supply. The variable resistor ($R_{v}$) values are specified in the context of the experimental sections.

In order to transform the response of the PD (in μA) to a signal which can be measured on an ADC channel of a microcontroller (ATMega328P, Arduino Nano), analog conditioning circuitry in the form of a transimpedance amplifier was used (see Figure 1). A large feedback resistance ($R_{f}$) in conjunction with an operational amplifier (Microchip, MCP 6002) was used to amplify the signal.

In the PEDD implementation, the anode of the detector LED was connected to GND and the cathode to one of the microcontroller’s standard I/O ports (PD2). The detection procedure follows the initial work on this topic [25], which is to say that the I/O was set to output, which charged the detector LED. This was then switched to input mode and the logic level was monitored via a counter over time, with the accumulated value being proportional to the discharge time and therefore the amount of incident light falling onto the LED. In order to bring temporal meaning to the counter, the microcontroller’s hardware timer (Timer 0) was used to count the number of microseconds passed when performing a measurement. Both the counter value and time per measurement were transmitted to a connected PC and recorded for processing at a later time.

**Figure 1 sensors-22-00254-f001:**
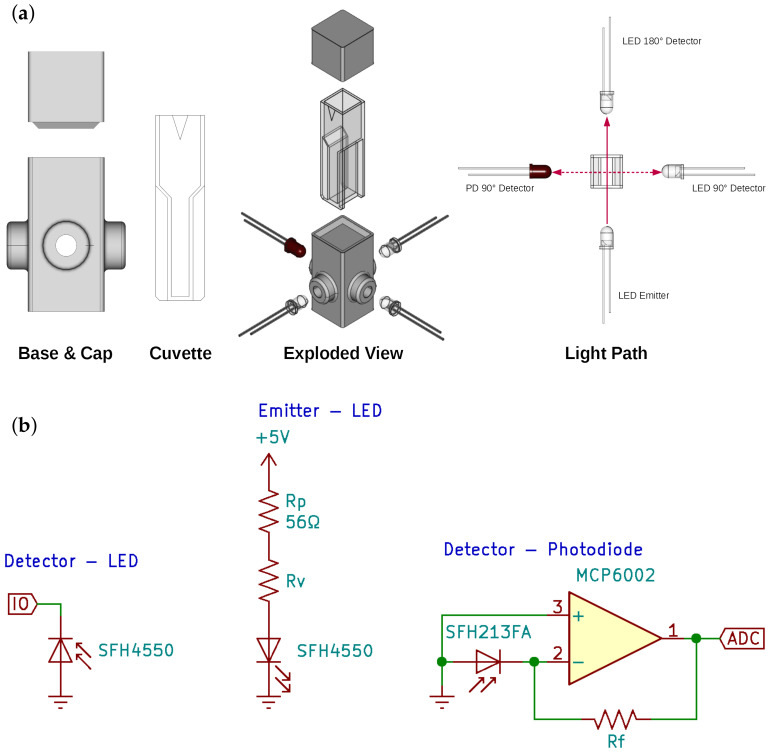
Chamber and electrical CAD design of the detection system. (**a**) Chamber CAD design and all possible light arrangements for the detector photodiode (PD) and light emitting diode (LED). (**b**) Electrical schematic showing the components and implementation required for the detector LED (**left**), emitter LED (**middle**) and detector photodiode (**right**). $R_{p}$: Protective resistor. $R_{v}$: Variable resistor. $R_{f}$: Feedback resistor.

### 2.5. Calibration Procedure

Measurement of the samples took place using the LED-PD and PEDD detector implementations. In both cases, two arrangements (facing and orthogonal to the emitter LED) were investigated for the turbidity standards by placing the detector and emitter as per Figure 1. Initially, the maximum and minimum concentrations were used for calibration, being placed in the holder, and $R_{v}$ was adjusted until the dynamic range of the response matched the measurement range via the respective microcontroller. Table 1 presents the $R_{v}$ and op amp feedback resistances of each set of conditions. For the photodiode, the emitter was set at its maximum intensity (Figure 1) and the gain established by varying the feedback resistor ($R_{f}$) on the transimpedance amplifier circuit. For the PEDD implementation, the microcontroller was programmed to measure at its maximum speed and monitored using Timer 0. As a result, the intensity of the emitter LED was varied until the response range lay within the 16-bit measurement software counter.

For the sensing implementations and arrangements, each suspension was measured for 60 s, with data streamed to the connected PC and recorded for later processing. Additionally, each measurement was repeated in triplicate in order to investigate reproducibility.

## 3. Results

### 3.1. Spectral Measurements of the Light Detectors

Figure 2 presents the EQE responses of the SFH 213 photodiode, SFH 213FA (with visible light filtering lens) and the SFH 4550 LED. This graph also features the sensitivity response of the LED as a detector, where it can be seen that under 860 nm ($\lambda_{max}$ of the emitter) illumination of the photodiodes converts incident light into current ∼10 times more efficiently than the LED does. This is further demonstrated with the current-voltage curves shown in the inset of Figure 2 in terms of a larger photo-current generation for the PDs and lower short circuit current of the LED. This is in line with our expectations and explains why the bulk of the literature continues to use photodiodes, rather than LEDs as light detectors, which is in agreement with earlier discussions [24,43]. Figure 2 also presents the normalised response of the LED along with the emission spectra of the excitation LED, which overlaps strongly with the response range of the LEDs being used as detectors. The spectral response of these LEDs is noted to be relatively narrow, ranging from 700–875 nm, contradicting Lau et al. who used LEDs as analytical detectors, stating that “*an LED is sensitive to all wavelengths of light equal to or shorter than the emission wavelength*” [25], although it is noted that they did not perform spectral measurements and this was likely an assumption they made. It is also possible that the poor sensitivity at short wavelengths is due to competitive light harvesting by the top contact. As LEDs are not primarily designed to operate as detectors, such factors are unlikely to impact the materials selection choices of LED manufacturers. The LED spectral photo response does, however, agree with Bui et al. [44], who observed the sensitivity band and the shift to shorter wavelengths by 40–60 nm when in detection mode, as these are typical Stokes shifts for inorganic semiconductors. The narrow response range ensured that this LED, as a detector, acts as a good band pass filter in the range of the emitter LED’s $\lambda_{max}$, which is particularly important for applications involving uncontrolled lighting conditions such as environmental monitoring in rivers or lakes. Here stray light could lead to erroneous readings such as false positives or negatives in either the orthogonal or facing arrangements.

### 3.2. Photodiode Detector (LED-PD)

Signals were collected from the PD directly facing the emitter and in the presence of the NTU standards (0–1000 NTU), as per Figure 1 and Table 1 (amplifier: $R_{f}=2.2\;k\Omega$) and are presented in Figure 3a. It can be seen that a good response is achieved which appears to possess an exponential relationship with the NTU, ($R^{2}=0.9995$). Initially, it was hoped that this configuration could be used to investigate the lower range (0–100 NTU) to provide higher resolution; however, this was not possible due to responses from the lower range going over scale when the signal gain was increased accordingly (increasing the $R_{f}$ on the transimpedance amplifier circuit). This is evident in the inset of Figure 3a, whereby the responses result in ADC values saturating as the $R_{f}$ is increased to 3.3 kΩ then to 5.6 kΩ. In order to more accurately measure the lower range, the measuring photodiode was placed in an orthogonal arrangement to the LED, with results of these experiments shown in Figure 3b. To achieve a more significant change in response, $R_{f}$ was increased to 10 MΩ, resulting in a dynamic range upper value close to the top of the measurement scale (5 V). An exponential growth model as fitted ($R^{2}=0.988$), with one data point, for 60 NTU, appeared to be an outlier, and may be attributed to experimental error. Regardless, this exponential trend highlights that the lower range results in a better response in the orthogonal arrangement. Examination of an even lower region (<10 NTU) using this arrangement may still not be suitable due to the sensitivity and uncertainty in this region, as highlighted by the error bars that represent one standard deviation.

While more complex signal conditioning approaches are possible, they would require the addition of more components to the circuit and thus cost; they are therefore considered beyond the scope of this study. This study is focussed on using a minimal number of components on the PD detector side for comparison to the LED detector. From this, it can be seen that the response in the figure is close to an optimum arrangement under this scope. Increasing the optical path length is another option for increased sensitivity, however this would impact miniaturisation and require a larger medium between the LED and PD path, which can increase the possibility of additional material and therefore lead to erroneous measurements. Overall, it appears that the LED-PD facing arrangement is very suitable for the 0–1000 NTU range and the orthogonal arrangement is suitable for the 0–100 NTU range.

### 3.3. LED Detector (PEDD)

Figure 4a presents the response obtained through the PEDD implementation, when the emitter and detector LEDs were facing each other. Here, due to the sensitivity of the detector LED, the emitter LED’s intensity was decreased by setting $R_{v}$ to 56 kΩ. Please note that this is an order of magnitude reduction in emission intensity compared to the LED-PD setup that used a total resistance of 56Ω. A very good fit ($R^{2}=0.998$) to an exponential growth model ‘$y=Ae^{x/\tau }+c$’ was observed for the relationship between LED discharge time and turbidity concentration. While the measurement of turbidity is ultimately time-based, a secondary y-axis is presented with the discharge count from a 16-bit software counter. This indicates the measurement resolution and it may be for this reason that the PEDD implementation was also capable of measuring the lower range (0–100 NTU) reliably—see inset. It is clear that this setup appears to be relevant for monitoring freshwater environments requiring both low (0–100 NTU) and high (0–1000 NTU) ranges.

Figure 4b presents the response of the LED detector with respect to turbidity concentration. The intensity of the emitter LED was increased by lowering $R_{v}$ to 330Ω, as less light falls on the LED in this orientation. It can be seen that the shape of the response can be approximated either by a power model ($y=A/{\left(1+ax\right)}^{-1/b}+c$) or to a lesser degree by two linear regions. For discussion purposes, the first linear region appears very sensitive at the lower range (0–100 NTU), also see Figure 4b inset. Upon visual inspection the sensitivity is estimated to be ∼400 Count_Discharge_/NTU and consists of over half the measurement range. In contrast, the larger range (200–1000 NTU) appears to have a sensitivity of 7 Count_Discharge_/NTU. Accounting for variation in the signal, see error bars (difficult to see due to occlusion by the markers and the reproducibility of the data), it appears that this implementation and arrangement may be applicable to the monitoring of drinking water, where levels of ≲1 NTU must be analysed. While such an investigation is outside the scope of this study, we strongly believe this warrants further investigation.

## 4. Discussion

### 4.1. Cost

Components and their power use directly impact the costs of remote turbidity sampling. When considering large scale deployments to meet monitoring requirements—an ethos of IoT systems—the component costs can scale accordingly with each required unit. Lau [25], Acharya [45] and Anh Bui [44] have all argued that LED detector systems are a lot cheaper when compared to a traditional LED-PD approach, both in terms of component cost as well as associated costs of the signal transduction circuitry. The LED-PD implementation requires analog circuitry including additional resistor(s), operational amplifier(s), an ADC channel and possibly also capacitors. With the exception of resistors and capacitors the necessary number of components for light detection and signal conditioning is listed in Table 2. With reference to Figure 1, the total major component cost difference (ignoring resistors) for a PEDD setup is ∼18% cheaper when compared to a LED-PD. While other studies in the literature agree that the PEDD arrangement will be cheaper, details pertaining to this price difference are often missing.

### 4.2. Sensitivity

In order to analytically compare the PD and LED, sensitivities and limits of detection (LOD) were established for each detector in each orientation. Despite the fact that the responses could be fitted to exponential functions, they were non-linear, meaning that sensitivities were approximated using piecewise optimisation by fitting two of linear models, e.g., see Figure 3b for the PEDD orthogonal arrangement. It can be seen that the linear fits are good ($R^{2}=0.987$), and as such it is considered to have two sensitivities. Table 3 lists the calculated sensitivities for the PD and LED detectors in both facing and orthogonal orientations. In general, it can be seen that the use of two-linear fits works well, with $R^{2}\geq 0.987$. Furthermore, the PEDD approach is confirmed to be more sensitive than the LED-PD approach, and follows the work of others in the literature who have obtained increased sensitivities for other analytes such as nitrite [46], transition metals [47], phosphate [29], pH [30], CO_2_ [31], and total urinary protein [48].

### 4.3. Limit of Detection (LOD)

Table 3 also presents the LOD of each data set, calculated based on Armbruster and Pry’s model when using data from a calibration plot [49]. It can be seen that the LOD achieved (using the LED detector) is lower than when using the PD by factors of 6.2 and 9.3, for the facing and orthogonal orientations, respectively. Note, that the drinking water has a requirement to have 0–1 NTU. Based on the calculated sensitivity (422.5) and LOD (0.898) the PEDD in the orthogonal orientation is a good candidate for monitoring drinking water. This could be further improved by decreasing the intensity of the emitting LED, as this would result in longer discharge times, improving the resolution in the lower NTU range. While this is outside the direct scope of this study, it is noted for future work.

### 4.4. Power

Table 3 lists the continuous power draw for each detection approach and orientation, based on the current it was observed to draw when powered by a regulated 5 V supply. In the LED-PD setup, the LED was driven at near its maximum rated power, and in conjunction with the operational amplifier the continuous power draw was ∼0.45 W. In contrast, the PEDD arrangement was more sensitive and as a result it required less power to drive the emitter LED. For the orthogonal arrangement the power draw was less than 0.02% of what was needed for the LED-PD system, while the facing arrangement was approximately an order of magnitude less again. With the increased sensitivity of the detector LED, it appears that it required less light and therefore a significant lower emission power draw. At such a power, a deployed system could run unattended for a much longer time than a comparable PD.

### 4.5. Other Considerations

Noise is an inherent factor when it comes to sensors. When comparing both LED-PD and PEDD approaches, the PEDD technique has achieved a significant higher sensitivity and lower LOD where noise is a considerable factor underpinning these characteristics, see Table 3. Based on these, the noise is significantly lower when adopting the PEDD approach. This is supported by one other study, i.e., by O’Toole et al. [29], who compared the LED-PD and PEDD approaches for phosphate determination. The reason for this is unclear without a more in depth study, however, it has been suggested that in the PEDD technique the measurement is spread over time, so the electronic noise is integrated and its effect is reduced [50], whereas in the LED-PD technique the measurement bandwidth is limited only by the Op Amp used in the transimpedance front-end of the PD. It must be noted that using an Op Amp in the LED-PD approach allows for temporal signal processing such as low/high/band pass filtering to account for noise and/or Fourier analysis—a process yet to be demonstrated using the PEDD approach. A recent study [51] investigated inherent noise/accuracy in the PEDD approach and identified temporal effects that may allow for analysis and filtering of EM interferences, thus allowing for a deeper comparison of both approaches in the presence of noise.

Another interesting discussion point arising from this study was the possibility of applying the PEDD using a photodiode in place of an LED. This was previously explored by Lau et al. [50], who compared five PDs and LEDs in PEDD mode. It was found that PDs were approximately 10 times more efficient at producing a photocurrent, which is in agreement with our observations in Figure 2. As a result the PDs discharged at a much faster rate than the LED counterparts and therefore the resolution of the PD was negatively impacted. This was also observed during this study, whereby the discharge rate of the PD was too great to register meaningful quantitative data. Overall, the lower photocurrent in LEDs appeared to be an advantage for the PEDD mode of operation and resulted in improved sensory characteristics, as discussed above.

## 5. Conclusions

This study reports the use of an LED as a detector for turbidity sensing, based on the ISO 7027 measurement standard. Specifically, the paired emitter detector diode (PEDD) charge/discharge technique was used and compared to a more typical photodiode LED setup, in both facing (transmission) and orthogonal (scattering) arrangements. The findings show that the PEDD approach required less components and was more cost effective (by ~18%), used significantly lower power (<1%), had a lower limit of detection (by factors of 6.2 and 9.3—depending on the configuration), and was substantially more sensitive (by a factor of ~117 for the 0–65 NTU range in the orthogonal arrangement). These improvements can have a significant positive impact on battery powered turbidity sensors, especially in terms of operational lifetime and cost.

## Figures and Tables

**Figure 2 sensors-22-00254-f002:**
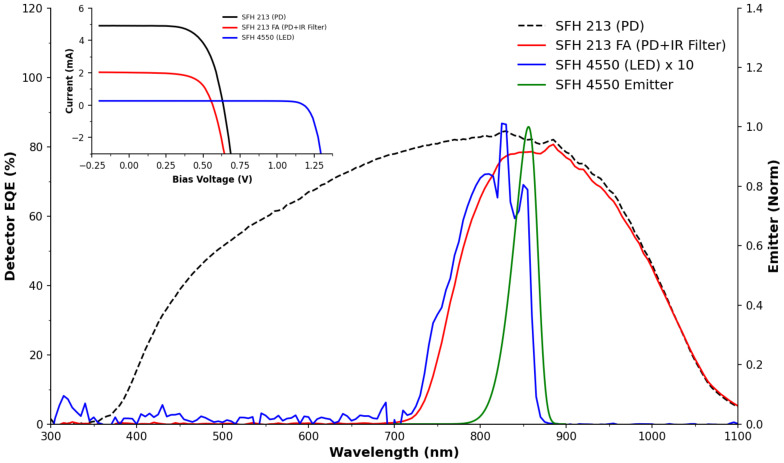
Spectral response of the photodiode (with and without the IR filter) and the LED in detector mode from 300–1000 nm. For reference the LED emission spectra model is shown ($R^{2}=0.9939$, normalised [0, 1]). Inset: Current-voltage responses of the PD and LED (detector) devices under simulated AM1.5 irradiation.

**Figure 3 sensors-22-00254-f003:**
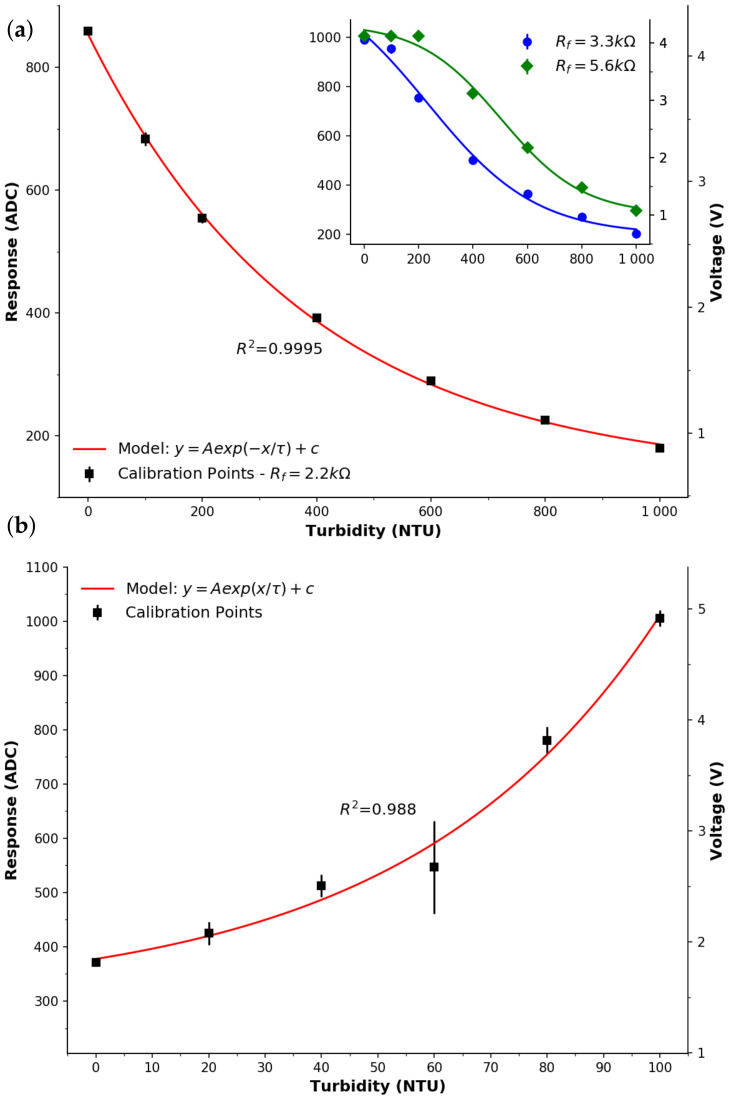
Response of the photodiode implementation. Points represent the average of successive measurements, error bars the standard deviation. (**a**) Photodiode facing the emitter ($R_{f}=$2.2 kΩ, ■). The line represents a good exponential fit ($R^{2}=0.9995$, $y=Ae^{-x/\tau }+c$). Inset—response plots with larger feedback resistors (● 3.3 kΩ, ⧫ 5.6 kΩ) on the op-amp, where the lines represent a sigmoid fit. (**b**) Photodiode placed orthogonal to the emitter. The line represents a good exponential fit ($R^{2}=0.988$, $y=Ae^{x/\tau }+c$).

**Figure 4 sensors-22-00254-f004:**
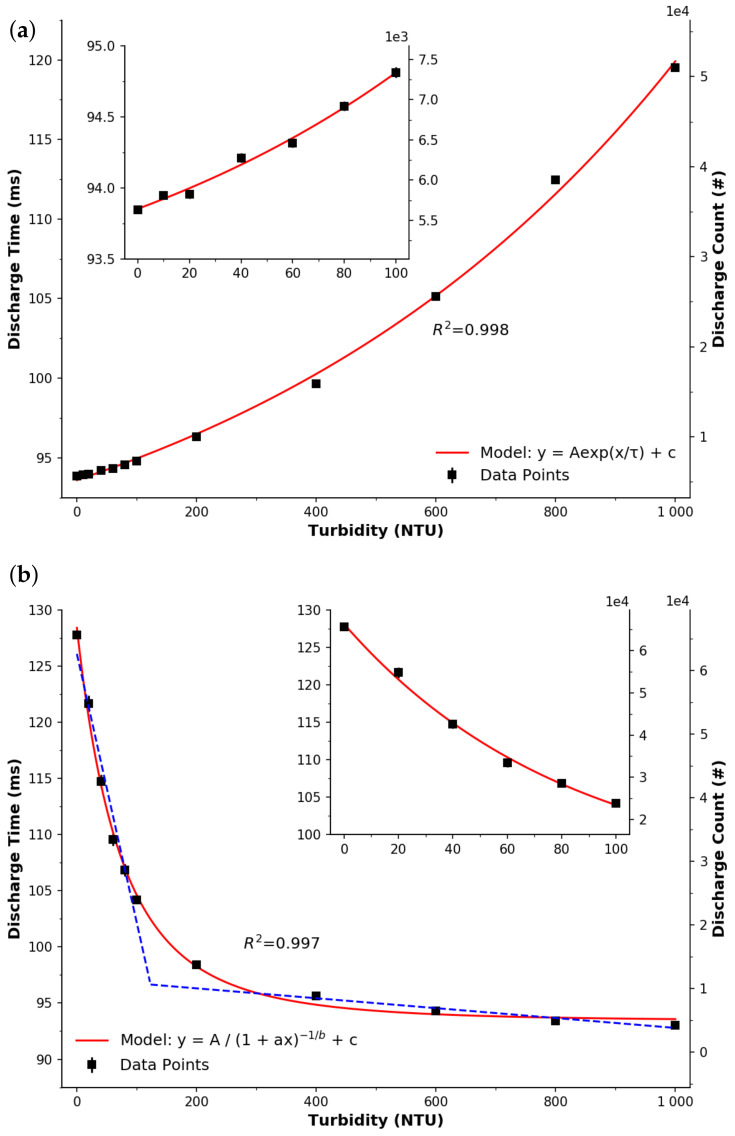
Response of the detector LED. The points represent the average of successive measurements and the error bars (difficult to see due to reproducibility of the data) represent the standard deviation. Insets—a magnified view of the response over a smaller range. (**a**) Detector LED facing the emitter LED. The red line represents a good exponential fit ($R^{2}=0.998$, n=12, $y=Ae^{x/\tau }+c$). (**b**) Orthogonal arrangement. The red line represents a good power fit ($R^{2}=0.997$, n=11, $y=A/{\left(1+ax\right)}^{-1/b}+c$). The blue dashed lines represent a dual linear fitted model ($R^{2}=0.987$).

**Table 1 sensors-22-00254-t001:** Implementation resistor values for each implementation and arrangement thereof.

Detector	Arrangement to Emitter	$R_{V}\;\left(\Omega \right)$	$R_{f}\;\left(\Omega \right)$
Photodiode	Facing	0	2.2 k
	Orthogonal	0	10 M
LED	Facing	56 k	N/A
	Orthogonal	330	N/A

**Table 2 sensors-22-00254-t002:** List of the major circuit components and their cost. Prices obtained from Digikey on 20 August 2020.

Component	Manufacturer	Part	Price ($USD)
LED	OSRAM	SFH 4550	1.12
PD	OSRAM	SFH 213	1.07
PD + IR Filter	OSRAM	SFH 213 FA	1.20
Op Amp	Microchip	MCP6002	0.42

**Table 3 sensors-22-00254-t003:** List of calculated sensitivities, ranges, and limits of detection for the four data sets.

Detector	Arrangement	$R^{2}$	Power	LOD (NTU)	Range (NTU)	Sensitivity (Unit/NTU)
PD	Facing	0.9923	447 mW	13.163	0–321	1.52
					321–1000	0.27
	Orthogonal	0.9971	447 mW	8.375	0–64	3.61
					64–100	11.47
LED	Facing	0.9996	446 nW	2.138	0–470	25.92
					470–1000	63.64
	Orthogonal	0.987	65 µW	0.898	0–123	422.53
					123–1000	7.78

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
