# Peer review of "Advances in Optical Based Turbidity Sensing Using LED Photometry (PEDD)"

_sensors, 2021, doi:10.3390/s22010254_

Round 1

Reviewer 1 Report

Well constructed article. 

The comments about Figure 3 should be improved (or the whole could also be removed).

Author Response

Pleas see attached PDF file.

Reviewer 2 Report

This paper reports the use of a LED as a detector for turbidity sensing, based on the 313 ISO 7027 measurement standard. The findings show that the PEDD approach was more cost effective (by ~18%) and used significantly lower power (< 1%) as well as having a lower limit of detection and was substantially more sensitive. These improvements can have a significant e impact on battery powered turbidity sensors, especially in terms of operational lifetime and cost.  Overall, this study provides a useful method to measure  turbidity, also, the manuscript is written well. Therefore, I recommend to receive the manuscript in present form.

Author Response

Pleas see attached PDF file.

Reviewer 3 Report

The authors present a comparison between the performance of two different approaches to the measurements of water turbidity. The paper is clear and the results are well documented, so, in my view, it deserves publication. In the following few observations/comments are reported.

1) Ref 11. “Schmitz, A. Low-cost Turbidity Sensors as a Method for Watershed Monitoring. Online” should be defined better: I was not able to find it online with a Google search.

2) Please define the acronym NTU, for sake of clarity.

3) Correct "Wavelangth" in the caption of Fig. 2.

4) In the PEDD technique a LED has been used as optical sensor: comments about the possible use of a PD instead of the LED as photodetector in the PEDD technique could be useful to the reader. A PD would be better or worse than the LED (for instance in terms of cost and sensitivity of the two photodetector to different portions of the spectrum) ?  

5) An interesting point is the impact of electronic noise in the two techniques: in the PEDD technique the measurement is spread over a large amount of time, so the electronic noise is in practice integrated and its effect is reduced, whereas in the LED-PD technique the measurement bandwidth is limited only by the OPAMP used in the transimpedance front-end of the PD. Is this an important factor to be taken into account in the comparison between the two approaches?

Author Response

Pleas see attached PDF file.
